# Creating an Optimal In Vivo Environment to Enhance Outcomes Using Cell Therapy to Repair/Regenerate Injured Tissues of the Musculoskeletal System

**DOI:** 10.3390/biomedicines10071570

**Published:** 2022-07-01

**Authors:** David A. Hart, Norimasa Nakamura

**Affiliations:** 1Department of Surgery, Faculty of Kinesiology, McCaig Institute for Bone & Joint Health, University of Calgary, Calgary, AB T2N 4N1, Canada; 2Bone & Joint Health Strategic Clinical Network, Alberta Health Services, Edmonton, AB T5J 3E4, Canada; 3Institute of Medical Science in Sport, Osaka Health Science University, 1-9-27 Tenma, Kita-ku, Osaka 530-0043, Japan; norimasa.nakamura@ohsu.ac.jp

**Keywords:** musculoskeletal repair, mesenchymal stem cells, inflammation, tissue engineering, tissue regeneration

## Abstract

Following most injuries to a musculoskeletal tissue which function in unique mechanical environments, an inflammatory response occurs to facilitate endogenous repair. This is a process that usually yields functionally inferior scar tissue. In the case of such injuries occurring in adults, the injury environment no longer expresses the anabolic processes that contributed to growth and maturation. An injury can also contribute to the development of a degenerative process, such as osteoarthritis. Over the past several years, researchers have attempted to use cellular therapies to enhance the repair and regeneration of injured tissues, including Platelet-rich Plasma and mesenchymal stem/medicinal signaling cells (MSC) from a variety of tissue sources, either as free MSC or incorporated into tissue engineered constructs, to facilitate regeneration of such damaged tissues. The use of free MSC can sometimes affect pain symptoms associated with conditions such as OA, but regeneration of damaged tissues has been challenging, particularly as some of these tissues have very complex structures. Therefore, implanting MSC or engineered constructs into an inflammatory environment in an adult may compromise the potential of the cells to facilitate regeneration, and neutralizing the inflammatory environment and enhancing the anabolic environment may be required for MSC-based interventions to fulfill their potential. Thus, success may depend on first eliminating negative influences (e.g., inflammation) in an environment, and secondly, implanting optimally cultured MSC or tissue engineered constructs into an anabolic environment to achieve the best outcomes. Furthermore, such interventions should be considered early rather than later on in a disease process, at a time when sufficient endogenous cells remain to serve as a template for repair and regeneration. This review discusses how the interface between inflammation and cell-based regeneration of damaged tissues may be at odds, and outlines approaches to improve outcomes. In addition, other variables that could contribute to the success of cell therapies are discussed. Thus, there may be a need to adopt a Precision Medicine approach to optimize tissue repair and regeneration following injury to these important tissues.

## 1. Introduction

### 1.1. Purpose of the Review

The purpose of this review is to present the current state of cellular therapy uses in enhancing the repair of injured or damaged tissues of the musculoskeletal system (MSK), and then propose approaches to improve the application of such approaches. The review focuses on soft tissues of the MSK that function in unique biomechanical environments. The organization is to initially present the scope of the problem and then address why inflammatory processes are relevant to the topic, and how they will likely influence outcomes of cellular therapy interventions. Subsequently, the discussion focuses on the way forward to improve outcomes regarding the healing of these tissues when injured. The content of the review and the perspectives presented are based on an assessment of articles in the PubMed database from the past 30 years with representative articles cited.

### 1.2. Characteristics of Musculoskeletal (MSK) Tissues

Tissues of the musculoskeletal system (MSK) (i.e., cartilage, menisci, ligaments, tendons, muscles, bone) are designed to provide a mechanical function to aid mobility. At the time of birth, many of them are “cell rich and matrix poor” but become more “matrix rich and cell poor” during growth and maturation discussed in [1,2]. This occurs as the mechanical demands increase and the cell density declines due to matrix deposition. In addition, many of these tissues have a low density of a microvasculature and innervation [3,4,5], with mature articular cartilage devoid of both. Some tissues, such as ligaments, also contain a few immune-related cells that could initiate an endogenous inflammatory response [6,7].

Many of the tissues of the MSK system have very complex organizations either at the macro or micro level. Tendons in different locations (i.e., Achilles tendon, patellar tendon, supraspinatus tendon of the shoulder) have different properties but all have an insertion site at the bone interface enthesis [8,9,10], a mid-substance and a myotendinous junction where it inserts into the muscle as discussed in [11,12]. The menisci of the knee have a central region devoid of a microvasculature and innervation that is collagen II-rich and a peripheral region that has a microvasculature and some innervation, discussed in [13]. The complexity of the matrix organization in the two areas are very different and are quite sophisticated in their complexity [14,15,16]. The template for the organization of this complexity is likely laid down during development and then expanded upon during growth and maturation. As this complexity was likely optimized during evolution to address the required function in the mechanical environment, it would be difficult to replicate many of them in a tissue-engineered construct. The exceptions to this conclusion are bone which heals quite well when injured in most locations, and muscle which also heals well in most circumstances.

Tissues of the MSK system are complex at multiple levels (macro structure and matrix organization) and vary in different mechanical environments. They are dynamic and can respond to changes in the mechanical environment. Bone can respond to changes in loading and become strengthened [17,18,19]. In contrast, several MSK tissues can undergo atrophy if not used appropriately and become “deconditioned” [20]. Following an injury, MSK tissues often requires immobilization (e.g., putting a leg in a cast when a bone is broken) which is a “deconditioning” environment for healing to initially take place.

This scenario poses some questions regarding signaling for healing and responsiveness to pro-healing signals/mediators. If one immobilizes the leg of a very young animal, such as a rabbit, the medial collateral ligament of the knee almost immediately ceases to grow in this anabolic environment [21]. The cells in what is normally a biomechanically active environment do not receive the signals regulating growth and growth stops in such a “deconditioning’ environment. After skeletal maturity, when the presence of anabolic mediators contributing to growth and maturation are diminished in expression, immobilization and deprivation of normal loading can lead to atrophy, potentially by the de-repression of a cassette of catabolic genes that includes some pro-inflammatory genes, such as IL-1 [22].

The characteristics of the various MSK tissues described above must be considered when attempting to repair or regenerate these tissues following injury.

### 1.3. The Inflammatory Response

Fundamentally, the inflammatory response is one that is designed to enhance survival after an injury (i.e., a cut in the skin) or exposure to environmental threats, such as microorganisms. After an injury to a tissue, the inflammatory response can also facilitate repair via clearance of damaged tissue components or microorganisms, and then initiate the fibrotic process with formation of a scar tissue, a process that is central to wound healing success as reviewed in [23]. Thus, scar formation in response to the injury of some tissues offers a survival advantage, but for MSK tissues that function in defined mechanical environments, scar tissue is not adequate for the optimal functioning of tissues, such as tendons, ligaments and menisci.

Because of its potential to also cause harm, inflammatory responses are regulated in a very complex manner, depending in part on the initiating factors, the extent of the response that is needed (i.e., the size and location of the wound) and the ability to initiate the downregulation of the response via both removal of the inciting events and elaboration of anti-inflammatory molecules, such as resolvins and related molecules [24,25,26,27]. In “normal” circumstances, the inflammatory response can be acute in nature followed by a resolution. However, if the inciting stimulus is not removed, this can lead to a state of chronic inflammation and fibrosis, with the development of pathology and loss of function of the affected tissues (i.e., pulmonary fibrosis, liver cirrhosis). After the transition from an acute inflammatory state to a more chronic state, such inflammation can become more difficult to control, possibly due in part to epigenetic alterations in the site of the response [28,29,30]. As some epigenetic alterations may be reversible [29,31,32], this may be an effective approach to enhance the potential for regeneration of damaged MSK tissues where a chronic inflammatory state is evident.

The timing of the onset of the ability to mount an inflammatory response during development has provided some interesting insights into the relationship(s) between tissue development and inflammation. Thus, the organization template of tissues and organs appears prior to an ability to mount an effective inflammatory response; however, some aspects of this relationship are still controversial as discussed in [23,33,34]. Injuries in some locations, such as cutaneous wounds, heal by regeneration if incurred early in fetal life, but they heal with the formation of scar tissue after the onset of effective inflammatory response capabilities are in place. Further study of scarless versus scar-forming wound healing may provide new clues to how to regulate the responses in adults to further attempts for successful tissue regeneration as discussed in [35].

Inflammatory responses are also influenced by sex steroids, and thus the response pattern would be influenced by both puberty in males and females, and in females after menopause, reviewed in [36,37]. It is well known that inflammatory processes can decline in the elderly, and this appears to involve estrogens [38,39].

## 2. Interactions between Inflammation and Injured Connective Tissues of the MSK System-Loss of Function

After skeletal maturity, injury to load bearing connective tissues, such as some ligaments, tendons and menisci can lead to the host attempting to initiate repair using the wound healing apparatus, or a failure to initiate such a response as in the case of complete rupture of the anterior cruciate ligament (ACL) of the knee, or after generation of a defect in articular cartilage. In the case of the ruptured ACL, the two ends of the ligament cannot find each other and thus a reconstruction with an autologous tissue section of a tendon [40,41] or an allogeneic tissue is required. In this circumstance, the reconstruction operation is another inflammatory stimulus with additional tissue damage due to the drilling of bone and cutting to gain access to the interior of the joint. In the case of articular cartilage defects due to the lack of innervation and a microvasculature, they are believed to contribute to a lack of healing unless some interventions are initiated, discussed in [42].

For those connective tissues that evoke an inflammatory response when injured, the result is the formation of scar tissue (outlined in Figure 1). In the case of the injured medial collateral ligament (MCL) of the knee, the injured tissue becomes rapidly “healed” with an early scar tissue that is comprised of a ~50–50 mixture of collagen II and collagen I [43,44] (the normal ligament is ~90% collagen). In contrast to the normal MCL, the collagen fibrils in the early scar tissue exhibit a random orientation that gradually becomes aligned along the length of the tissue over time [43]. However, even when aligned, the collagen fibrils are of a small diameter compared to the biphasic size distribution in the normal tissue. By two years post-injury, some large collagen fibrils appear but the ligament is much weaker than normal for much of this time period [44]. For the MCL, this weakness does not compromise function as it is a stabilizing ligament that operates normally in the toe region of its stress–strain curve. However, this model does show that in the context of an inflammatory response, the healing process does not lead to a regeneration of the normal tissue even after a protracted period of time post-injury or post-surgery. The process does, however, lead to a partially functional tissue.

The situation in a reconstructed ACL is somewhat different, but again shows that an inflammatory process can likely lead to a functional compromise of the tissue. When implanted, a patellar tendon or hamstring tendon graft is likely stronger than the original ACL but has some properties that differ from the ACL as discussed in [45]. Using an allogeneic ACL graft should provide at least something of equivalent strength. However, over time, such reconstructions begin to undergo creep and stretch out [46,47,48], with the implanted material becoming more scar-like. The basis for this response pattern could be in part, due to damage to the grafts while preparing for the reconstruction, and/or due to the inflammation associated with the operation contributing to a local inflammatory environment that persists. Interestingly, treatment of the graft environment with an anti-inflammatory glucocorticoid can prevent or diminish this effect [49,50]. In contrast to the MCL, the ACL operates in a high stress environment and so the functional compromise over time that is observed following reconstruction can lead to dysfunction of the knee as discussed in [46].

The consequences following an injury to the menisci of the knee are likely more complex than those for the ligament injuries. The mechanical environment for a meniscus is complex, with the central area exposed to compressive loads, while the periphery is more ligament-like and is subjected to hoop stresses, discussed in [16]. Injuries to the periphery often do heal, depending on the extent and type of injury, perhaps due to its nascent blood supply and innervation. However, being in a high stress environment often leads to repair failure, and if the tissue has to be surgically removed, it can lead to an increased risk for OA as reviewed in [51]. Of relevance to this discussion is the fact that some injuries to menisci of the knee in young individuals do heal with some interventions [52], likely via the more anabolic environment operative during active growth and maturation.

Injuries to articular cartilage do not heal, and even surgical interventions to promote repair either do not work well or are only temporary in outcome. Initiation of surgical interventions to stimulate repair by microfracture of the bone beneath the cartilage leads to fibrocartilage and not hyaline cartilage [53,54], and if left alone, many defects will progress to overt OA over time. Transplantation of chondrocytes from non-loading parts of the joint to defects arising in highly loaded areas of the cartilage do offer some repair potential as reviewed in [55,56].

The above discussion leads to two important conclusions: (1) inflammation associated with injury or surgical interventions to mechanically active connective tissues of the MSK system serve an important function in relation to scar formation, but need to be controlled as scar tissue compromises the function of many of these tissues; and (2) a return to function of injured connective tissues in mechanically active environments is often compromised and new approaches are needed to facilitate regeneration and return to function (outlined in Figure 1). However, even the use of newer interventions with tissue engineered constructs to regenerate compromised tissues will need to consider effective control of inflammatory processes which could compromise the effectiveness of such approaches over the long term.

## 3. Factors That Can Complicate Post-Injury Processes and Inflammation

The repair of connective tissues of the MSK system damaged by injury and/or disease, particularly in skeletally mature subject or older individuals where the natural healing process has been compromised during the aging process, may be an influencing factor. As individuals age, the immune and inflammatory processes can diminish, as reviewed in [36]. In females, inflammatory responses may be altered after menopause, possibly via the effects of loss of estrogen on macrophage functioning [38,39]. Inflammatory responses in females can be altered during pregnancy as discussed in [57,58]. Therefore, females may generally regulate inflammatory processes differently than males. Therefore, sex and stage of life are important variables to be aware of in planned studies.

While normal healthy young adults can usually heal without complications, there are factors that can potentially interfere with healing processes in addition to those mentioned above. These include the presence of diseases, such as diabetes, inflammatory autoimmune diseases and obesity, reviewed in [59,60]. In a preclinical rat model, the presence of induced type 2 diabetes led to altered healing of an injury to the Achilles tendon [61,62,63], and it is well known that, in humans, those that are diabetics often do not heal well or exhibit delayed healing, reviewed in [60]. Obesity can lead to development of metabolic syndrome, with an on-going low level of inflammation, reviewed in [64]. Obesity can also lead to altered structure of tendons, such as the Achilles tendon [65,66], which may increase risk of injury. Based on these considerations, co-morbidities can likely also impact the healing of injured connective tissues, such as tendons, reviewed in [60], and others.

Obviously, the use of some medications to treat co-morbidities could also influence outcomes depending on the type and dosages being used. Treatment with anti-inflammatory medications, such as NSAIDS [67,68,69] or high or continuous doses of glucocorticoids [70,71], could potentially adversely affect outcomes after interventions with cell therapy approaches. Therefore, the presence of co-morbidities and their treatment, as well age and sex, should be recognized and addressed before initiating interventions to enhance the repair/regeneration of injured connective tissues of the MSK system. Failure to do so could complicate the results of clinical trials using experimental interventions, such as cellular therapies.

## 4. The Role of Mechanics in Connective Tissue Repair and Regeneration: The Interface with Biology Is Critical

While many tissues function in the context of a mechanical environment, such as the lung, skin, and the cardiovasculature system, tissues of the musculoskeletal system function in a variety of compressive, shear or tensile environments, and their complex structures reflect such requirements in unique and specialized environments. While the composition and functioning of tissues, such as tendons, ligaments, menisci and articular cartilage can undergo changes with aging [2,72,73,74], it is not known in detail if they have evolved to their optimum to last for >80–90 years. It is clear that structure and function relationships are critical to the performance of functional activities. The prevailing wisdom is that engineering of artificial MSK tissues should lead to a construct that mimics the original that developed in utero and during growth and maturation, discussed in [75].

There are various approaches to address the issue of the biomechanical environment with regard to the development of constructs to facilitate the repair and regeneration of damaged or diseased MSK tissues. The first is to use an artificial scaffold containing cells, with the scaffold supplying a somewhat rigid but biodegradable template after implantation to allow the cells to adapt to the in vivo conditions in a mechanically active environment [76,77]. A second approach is to condition the in vitro generated scaffold containing matrix molecules, such as collagen I to which cells have been added, in a biomechanically active environment in vitro, prior to implantation [78]. These two approaches allow for the cells in the constructs to adapt to early loading and respond with enhanced secretion of extracellular matrix components, as well as adapt to loading and make adjustments to the cellular apparatus to allow survival and functioning when implanted. While some of the studies have used cells derived from tissues, others have shown that undifferentiated mesenchymal stem cells can also respond to mechanical loading in vitro in unique manners [79,80,81,82].

In studies with a tissue engineered construct [TEC] containing synovium-derived MSC and the matrix generated by these cells in culture as discussed in [83], the TEC were generated in the absence of loading in vitro. Following implantation, the cells appear to respond to the in vivo loading environment and form a hyaline-like repair tissue based on in vivo cues and the cells that are presented [84]. In this model, it would likely not be advantageous to expose the cells to mechanical loading regimens in vivo as the self-aggregation of the in vitro generated TEC is critical for the subsequent in vivo implantation. Exposure to in vitro loading may interfere with the post-implantation process. For other applications focused on the repair of MSK tissues functioning in tensile-loading environments, in vitro loading could be beneficial.

While there is some variation in the use of mechanical loading of constructs for the repair and regeneration of damaged MSK tissues, their use depends on the type of tissue being repaired (i.e., ligament, tendon or cartilage) and the in vivo loading environment that the constructs will be subjected to following implantation. In some circumstances, in vitro loading prior to implantation could lead to the generation of a construct with a better organized matrix and increased mechanical integrity, but in other situations, such loading could compromise some of the attributes of the construct. Therefore, the use of in vitro loading depends on the applications a construct will be used for in vivo.

## 5. Enhancing Repair/Regeneration of Injured Connective Tissues of the MSK System

Given the limitations or variables affecting healing after an injury, much research has focused on developing new approaches to enhance repair, often with the goal of tissue regeneration. One approach involves the use of Platelet-rich Plasma (PRP) [85,86,87], while others have used growth factors [88] and molecular blocking approaches (anti-sense, specific antibodies, enzyme inhibitors) as reviewed in [89,90,91,92,93].

PRP is usually derived from autologous blood and then injected into the site of a wound such as that in a tendon [94,95], ligament [96,97,98] or meniscus [99]. As platelets contain a number of growth factors and other relevant molecules, by injecting the PRP into the wound site at the time of surgical repair after degranulation the platelets can release their contents and create an enhanced anabolic environment. Such preparations have also been used in the conservative treatment of OA of the knee where the PRP is believed to alleviate the symptoms of pain and perhaps exert an anti-inflammatory effect in the intra-articular space reviewed in [85,86,87]. Unfortunately, in the latter scenario the efficacy of the PRP can be variable and not all patients respond positively. Whether this variability is due to the platelets and their content, or the method of preparation is not well defined. There are several methods that have been used for the preparation of PRP [85] as well as host variables that could impact the efficacy (e.g., co-morbidities, medications,). Some applications of PRP for the treatment of MSK diseases, tissues and conditions are summarized in Table 1. As this is a large literature, the citations in Table 1 are representative of the field in general.

As an alternative to PRP as a source of growth factors and other anabolic molecules, some studies have used specific growth factors as supplements in an attempt to enhance the healing of injuries to these connective tissues [116,117,118]. Thus, some studies have used growth factors, such as angiogenic factors [119] and others, such as IGF-1 [118,120], in an attempt to enhance healing. One of the limitations of this approach is the short half-life of growth factors or the binding of growth factors to extracellular matrix (ECM) components, which makes them unavailable for interacting with cells.

Use of approaches that can lead to blockages of specific steps or molecules during healing has also been tried by many investigators, but with limited success as reviewed in [90,91,93,121]. Some of the limitations relate to the half-life of the molecules, the specificity of the interventions or a failure to disseminate throughout a dense scar tissue, even when early in the process [121].

It should be noted that most, if not all, of the previously described studies did not attempt to control any on-going inflammatory processes that were occurring in conjunction with the injury or any surgical intervention. The use of anti-inflammatory drugs immediately after induction of the injury and/or a surgical intervention is shown to inhibit sequelae to an injury or the inflammation in a joint. This is shown in both rabbits [49] and pig models [50]. However, one would have to be careful not to use concentrations of such anti-inflammatory molecules so as to not interfere with the normal healing process. Furthermore, high doses of drugs, such as high dose glucocorticoids, may have unintended consequences regarding mesenchymal stem cells in a joint that could participate in the healing process [70].

Thus, going forward, if attempts are made to improve the healing environment to achieve better healing outcomes, an approach using both anti-catabolic/inflammatory elements and a pro-anabolic aspect should be utilized in order to optimize their potential.

## 6. Use of Single Cell Preparations of Mesenchymal Stem/Signaling Cells (MSC) to Enhance the Repair/Regeneration of Damaged Mechanically Active Connective Tissues

Cells called mesenchymal stem cells, mesenchymal stromal cells or mesenchymal progenitor cells have been studied primarily since the early 1990s, reviewed in [122,123]. These cells express a subset of cell surface antigens and can be induced to differentiate in vitro into cells of various lineages, such as osteogenic, adipogenic and chondrogenic cell lineages. They can be isolated from bone marrow, adipose tissue, skin, brain and many tissues reviewed in [123]. Differences in the ability to differentiate towards the different lineages were noted between cells isolated from different locations [122,123,124]. However, MSC isolated from individual tissues demonstrates extensive heterogeneity, discussed in [123,124,125].

While the cells that were labeled mesenchymal stem cells ~30 years ago, attempts to use preparations of free MSC to repair damaged connective tissues by the injection of millions of cells into a local space, such as a joint, or systemically into the circulation, they have not yielded a reproducible effective repair of the damaged tissues, possibly due to a failure to home and be retained at the injury sites [108]. Some of this limitation may be overcome by engineering membrane expression of molecules to enhance localization, reviewed in [123]. While the injection of free MSC into osteoarthritic knees did not lead to overt repair of the damaged cartilage, it was noted that injection of such cells from various sources could lead to a lessening of the pain and inflammation of OA, reviewed in [126,127]. Some studies indicate that MSC were injected, but many reports used preparations that were not pure MSC and instead were a mixture of cells labeled Bone Marrow Aspirate Concentrate (BMAC) that may contain BM, MSC or BM stromal cells, but also other cells from the BM discussed in [87,128]. A recent report [129] indicated that BMAC was more efficacious than PRP, but it cannot be concluded that this was due to the MSC in the preparations.

This failure of free MSC to initiate effective repair of damaged tissues led Caplan to hypothesize that perhaps MSC should be relabeled Medicinal Signaling Cells (MSC) as they may function by secreting or releasing vesicles containing factors or mediators that enhance the ability of endogenous cells to initiate effective repair [130]. In this scenario, MSC would release factors that would interact with residual endogenous cells at a site of injury to then repair their own tissue. While this is an interesting possibility, such abilities may be compromised by an inflammatory environment at the site of tissue injury [131,132]. Thus, in the intra-articular environment is an inflammatory process which led to an alteration of the synovial fluid MSC that compromised their ability to aggregate, likely via the actions of the mediator MCP-1 [131]. Therefore, unless such inflammation is curtailed, the MSC might still be compromised in fulfilling a role as a signaling cell,

Furthermore, if indeed MSC are signaling cells, they should likely not be used as a treatment of “last resort” in situations like OA when the articular cartilage is in a severely degenerated state so there may be little template available to initiate effective repair. With regard to other injured tissues, such as tendons and ligaments, it remains to be determined whether the addition of MSC treatments will help “re-direct” early scar-forming cells towards a more normal tendon or ligament structure. Of note is that once injured, there is also an initial inflammatory environment, but also there is a loss of biomechanical integrity and thus the initial scar tissue is not biomechanically loaded in a real functional manner, and it takes some time for the collagen fibrils of a scar to realign to allow for function loading again [43]. How the MSC would interpret the direction of need in such circumstances remains to be determined.

While the concept of MSC with a signaling role may still have some potential limitations, there are recent lines of evidence that indicate that MSC can release extracellular vesicles (EV), sometimes labeled exosomes, that are membrane enclosed packets containing growth regulators, miRNAs, and other relevant molecules as reviewed in [133,134,135,136]. In a repair context, once MSC are localized they could release EV-containing molecules that could enhance the endogenous healing process, as these EV could then be taken up by endogenous cells leading to enhanced healing. Thus, the molecules contained in EV would be somewhat protected from degradation that might occur if they were secreted as individual molecules, particularly by proteinases or RNases in an inflammatory environment. However, while EV may enhance healing under controlled conditions, it remains to be determined how effective they may be when injected in vivo into inflammatory conditions.

In addition, it also remains to be determined whether the differentiation potential of MSC as progenitor or stem cells should be dismissed in favor of strictly a signaling role, discussed in [125]. Certainly, both roles may be useful in the enhanced repair of tissues when used in a tissue engineering approach to generate constructs that appear to enhance the healing of human cartilage defects when implanted [137]. Some applications of MSC for the treatment of MSK diseases and conditions for specific tissues are summarized in Table 2. As this is a large literature, the examples indicated are representative of this large and expanding field.

## 7. Use of MSC in Tissue Engineered Constructs to Enhance Repair of Injured or Diseased Tissues

While the use of free MSC has not yielded consistent success in repairing damaged tissues, using them in constructs generated in vitro has led to some successes for some tissues. MSC isolated from a variety of tissues have been isolated and then often incorporated into synthetic scaffolds, scaffolds with other ECM-like matrix components, an endogenous natural protein matrix or a hybrid synthetic/natural matrix, reviewed in [156,157,158,159]. Recent advances in bioprinting may offer more sophisticated and complex scaffold-cell constructs [160,161,162]. Such constructs are then implanted into defects in tissues or in an injury site. As many of the scaffolds used are biodegradable, it is then hoped that they will be replaced by a natural matrix over time. Interestingly, in nearly all reports of studies assessing the efficacy of such implants, there is no mention that the inflammatory environment generated by the implantation procedure has been controlled or addressed in any manner. In spite of this limitation, some successes are reported, particularly in the repair of articular cartilage which does not heal effectively without intervention.

While many studies have reported the use of MSC in biodegradable scaffolds, several studies using a Tissue-Engineered Construct (TEC) consisting of synovium-derived MSC in a matrix secreted by these cells in vitro for the repair and regeneration of difficult to repair tissues, such as articular cartilage [83,163], menisci [164,165,166] and intravertebral discs [167]. The initial cartilage studies were performed in a large animal model (pigs [83,163]), while more recent studies have resulted from implantation into patients with articular cartilage defects in a pilot “proof of principle” design [137,156]. The advantages of the autologous TEC approach are: (1) it does not require an artificial scaffold as the matrix generated by the in vitro culturing serves that purpose; (2) once released from the culture dish, it spontaneously aggregates into a construct with the cells and matrix intermixed; (3) when implanted into an injury site it adheres to the residual tissue and does not require fixation, possibly due to the fibronectin in the construct; (4) the cells in the TEC are not differentiated prior to implantation but then appear to differentiate in vivo and respond to local environmental factors including the mechanical loading conditions, or interact with endogenous cells to facilitate repair. In the case of articular cartilage repair, after implantation, the TEC leads to development of the layered structure of articular cartilage with a change in matrix molecules production appropriate for the in vivo conditions [84,165]. In the pilot studies in patients with articular cartilage defects, the implanted tissues have been assessed post-implantation [122,123]. While there is some variation in the structures resulting from the TEC implantation, all defects were filled, and some appeared to be regenerated [137,156].

It should be noted that the TEC implantation studies in both the preclinical models and the pilot human studies, the authors did not attempt to control any inflammation resulting from the implantation surgery or the initial event leading to the formation of the defects. In addition, they did not add any potential anabolic stimuli, such as PRP, to possibly negate the inflammation and provide further enhancement of healing. These latter points are interesting since all of the patients were skeletally mature (both males and females) and thus the post-implantation differentiation occurred in the absence of any factors present during early growth and maturation prior to puberty. As these studies were focused on limited defects in the articular cartilage, there was likely sufficient residual cartilage present to serve as a template and/or provide local factors required to maintain cartilage which were active when provided the right cell/matrix construct. It remains to be determined in detail how a local mechanical environment contributes to the development of articular cartilage, menisci, or intravertebral discs in conjunction with biological cues, discussed in [75].

While the studies in both preclinical models and in the pilot studies with patients with articular cartilage defects have exhibited very promising results, the outcomes are likely still in need of improvement. It was noted in the porcine studies that the repair cartilage exhibits hyaline-like characteristics but does not lead to regeneration of the surface layer [168] that has been called the lamina splendens [169]. As this superficial surface layer has been suggested to serve a barrier function for the hyaline cartilage [170] as well as a lubrication function [171], failure to regenerate this barrier may compromise long term survival of the repair tissue, reviewed in [172,173,174]. This conclusion is supported by the studies of Takada et al. [170] who reported that if the lamina splendens is disrupted, materials can gain entrance or exit from the cartilage, and it predisposes the development of OA. In rats, it appears that the lamina splendens arises during early post-natal life [170]; but whether it arises with a similar timeframe in humans could not be found. However, it has been found in other species discussed in [172]. Furthermore, what the stimulus is for the development of a lamina splendens is also not known, but it could relate to the biomechanical environment and the presence of unique growth and differentiation regulators in early post-natal life. However, the importance of the lamina splendens has led to some research effort to synthesize an artificial structure which could serve some of the functions of the lamina splendens [175]. This is an area for future research, given the lubrication and barrier function of the lamina splendens.

Additionally, regarding the presence of endogenous growth regulators, in the human TEC studies [137,156], the patients ranged in age from 28 to 46 years old and were therefore somewhat young. Preliminary preclinical studies in the porcine model demonstrate that the TEC approach is equally effective in skeletally immature and mature pigs [83], but studies were not performed with older pigs. Thus, it remains to be determined whether there is any age-dependent decline in the effectiveness of the TEC approach that may be attributed to an age-related decline in growth regulators in the tissues, or even after menopause in female patients. If such a decline is observed, it may be overcome with the use of autologous PRP [85,86], as long as the platelets in older individuals have not also been compromised.

With the preferred use of autologous MSC by both patients and some regulatory agencies, the use of MSC from young versus older/elderly patients is also a consideration as MSC numbers and function in some available depots also decline with age [176] and there is the potential for perhaps epigenetic alterations due to life experiences or exposure to chemicals that could potentially contribute to the compromised function [177]. To overcome this potential limitation of autologous MSC, some parents are having their children’s cord blood MSC frozen in case they are needed, or the use of standardized allogeneic MSC from a source, such as an expanded cord blood MSC, has been proposed [178,179]. Thus, optimization of conditions to achieve the best success may require both the most appropriate MSC and the best in vivo conditions that can be obtained.

## 8. The Way Forward

While progress has been made toward using cell therapies to improve healing outcomes, there is still a need for further improvements. As discussed above, the attention has been focused on the implantation of cells, such as MSC, rather than trying to optimize the in vivo implantation environment. Likely, the way forward will require addressing more attention towards optimizing both what is implanted and the environment that it is implanted into. Clearly, attention to such variables will be complex and it is very likely that “one size does not fit all”.

The adverse effect of inflammatory processes on the repair and regeneration potential of cellular therapies is of central concern, and such processes will have to be controlled if expectations of further success regarding cellular therapies are to be achieved. While the successes achieved thus far have provided support for further investment and research, some of the diseases or conditions that could benefit from cellular therapies will likely be more complex and challenging.

Using articular cartilage repair/regeneration as an important example due to the current successes and the need as cartilage-related conditions, such as osteoarthritis, affects tens of millions around the world, and with alternatives to cell therapies (i.e., drugs, exercise, injury prevention) of limited impact thus far. Osteoarthritis is both a disease of mechanics [180] and inflammation [181], and of the whole joint and not just the articular cartilage [182,183]. The term osteoarthritis is actually an umbrella term that encompasses several subtypes of OA including post-traumatic OA, metabolic OA associated with obesity reviewed in [20,42,181,184], post-menopausal onset OA discussed in [42], with idiopathic OA a large subpopulation of patients for which a link to a cause has not been clearly defined. Thus, OA, which can develop if a cartilage defect is not repaired, is heterogeneous and complex, and one cell therapy solution likely will not apply to all subtypes of the disease.

While the transition from repairing fresh cartilage defects using cell therapies may identify the need to develop several different lines of approach, there are likely some principles that the approaches should share. The first is that, unless the biomechanical environment is restored or the biomechanical compromise addressed, any cell therapy may not achieve long-term restoration of function. Second, as most OA patients are older and many will have co-morbidities, any cell therapy intervention will likely require a co-intervention to augment the need for an anabolic supplementation, such as PRP or EV. Thirdly, and relevant to point two, there will be a need to control inflammation, often in the context of diabetes, which can contribute to an altered inflammation [60]. In addition, rather than an acute inflammation as perhaps with a cartilage defect, those with OA may have converted to a chronic form which may require different strategies. Fourthly, the cell therapy cannot be considered the intervention of last resort when most of the articular cartilage is degraded, and the disease is considered end-stage. At this point there is little cartilage template remaining for a cellular therapy to enlist in the repair/regeneration effort. As the MSC in a TEC may not only contribute to the repair of the tissue damage by replacement, the undifferentiated MSC in the TEC can also release EV that could travel to remaining chondrocytes in the residual cartilage to contribute to the repair effort. Thus, repair via cell therapy interventions should likely be initiated early in the disease process rather than later.

While the above discussion has focused on repair of articular cartilage, some of the principles discussed can also be applied to the healing of other connective tissues of the MSK system, including menisci, intravertebral discs (IVD), as well as ligaments and tendons. In the case of tendons, tendons in different locations exhibit different properties [185], tendon properties can change with aging [74,186,187] and some tendons, such as the supraspinatus, can undergo age-related degeneration without overt symptoms [188,189,190]. Thus, cell therapy treatment could be envisioned to address tendinopathies rather than overt ruptures. Similar issues can likely also be applied to other tissues, such as menisci and IVD.

Another separate set of variables that could affect the efficacy of cellular therapies is genetics and epigenetics. Genetics and epigenetics could affect the MSCs, with age-related epigenetic changes potentially affecting the functionality of the MSC later in life when they are needed for cell therapies [191,192,193]. Genetics and epigenetics could also affect the target tissues of the cellular therapies. For example, some individuals with Marfan’s Syndrome or the spectrum of Ehlers–Danlos Syndrome [194] may appear to have mutations in some of their ECM proteins that impair function and increase risk for injury or tissue failure reviewed in [193,195,196,197]. Thus, the outcome of the cellular therapy may not be optimal when using autologous cells and allogeneic cells may be preferred [198] or correcting the MSC via in vitro alterations [195]. While the examples presented are rare, there may also less overt variation in connective tissue molecules that predispose to injury or poor healing that do not present with symptoms, and these could also influence outcomes of cell therapies.

As the use of cellular therapies including the use of MSC continues to expand into more complex disease scenarios, it is clear that the use of multiple modalities in addition to the MSC will be needed. Some of the variables may be more readily assessed, but as continued improvement in genetic analysis and characterization of the epigenome become more common place, an element of precision medicine will be applied to the use of complex cellular therapies for MSK connective tissue repair and regeneration.

## 9. Conclusions

Attempts to enhance the repair and regeneration of injured connective tissues of the musculoskeletal system using cell therapies has been the subject of intense research over the past 30+ years. With the discovery of cells with the ability to differentiate into several relevant lineages (e.g., chondrocytes, bone cells, and others) reviewed in [122], this effort intensified. Using cells labeled mesenchymal stem cells (MSC), expectations ran high, but achieving success was more challenging.

Early after the discovery of adult “stem or progenitor” cells, there was considerable anticipation that they would rapidly be used to repair a variety of tissues damaged by injury or disease, particularly during the aging process. This hope rapidly became hype, and a number of what have been called rogue clinics and companies began selling stem cell-based cell therapy approaches directly to patients or consumers for a number of conditions [199,200,201]. Such rogue entities preyed on desperate patients, and many such clinics in North America have been recently curtailed by the FDA and Health Canada in the USA and Canada, respectively. Such rogue applications of stem cell therapies emphasize the need to continue to develop methods and interventions to use these cells more effectively and with a solid base of scientific and clinical justification. This will require building on past successes and failures to evaluate new directions and approaches.

Learning from these past scientifically and ethically approved research efforts, it is emerging that many of the relevant connective tissues that could benefit from stem cell interventions have complex structures, are designed to work in complex mechanical environments, and when injured this creates an inflammatory environment. Furthermore, when injured or subjected to a disease process, the situation arises as an adult or an elderly individual when the anabolic environment of youth (growth and maturation) is no longer evident. Thus, attention to the environment that cells, such as MSC, are placed in, either as individual cells or incorporated into constructs, needs to be addressed if the MSC are to achieve more of their potential to impact the return of functionality in these connective tissues. That is, control of an environment where a catabolic inflammatory process is needed, supplementation of the environment with appropriate anabolic mediators is also needed (either as molecules, PRP or extracellular vesicles), and for some circumstances using cellular therapy early in a disease process while the remaining endogenous tissue can serve a template function may additionally be critical. Finally, controlling the impact of co-morbidities (i.e., diabetes) may also be required. Thus, improving the environment into which the cells are placed may be critical for further success. Similarly, picking the right cells for the job may also be critical as MSC from different sources can exhibit different properties even though they can have a similar phenotype, as discussed in [123,124]. Thus, the right cells in the right environment at the right time are needed are discussed in [202,203,204], and there is likely a need for a more “precision medicine” approach as “one size does not fit all” [202].

While some progress is being made in the applications of cellular therapy, including MSC use in tissue engineered constructs as reviewed in [165,198], and many lessons have been learned as outlined above, several questions related to the issue of tissue regeneration still remain. The first relates to human heterogeneity and how such heterogeneity translates to variation in connective tissue structure and function. A second relates to the question of whether absolute regeneration is required to obtain optimal functioning in a specific mechanical environment? That is, would 80 or 90% regeneration at a structural level be sufficient for people in the 60–70 years old age range, but perhaps not acceptable for someone 30–40 years of age and wanting to maintain a very active lifestyle? Some of these philosophical questions may also need to be factored into the expectations of how success is defined going forward.

## Figures and Tables

**Figure 1 biomedicines-10-01570-f001:**
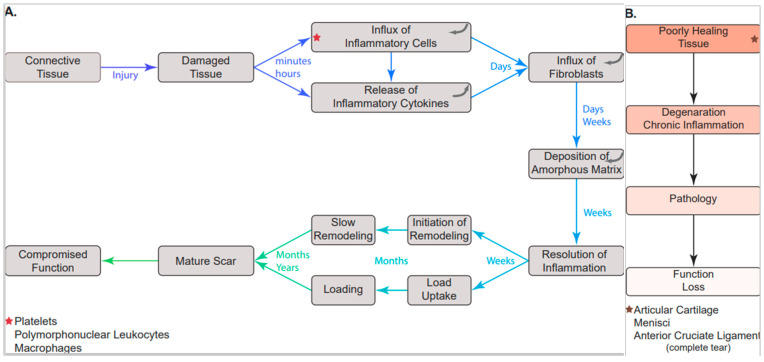
Sequence of events following injury to soft tissues of the MSK system. Following injury to most soft tissues of the MSK system, a sequence of events leading to scar tissue formation and maturation occurs (**A**). This sequence of events involves an inflammatory response that resolves over time as the scar tissue forms and matures. In some tissues, these events do not lead to a healing response and involve an inflammatory response that may become chronic with development of pathology and loss of function (**B**).

**Table 1 biomedicines-10-01570-t001:** Applications of Platelet-Rich Plasma in Cell Therapy for Connective Tissue Repair and Regeneration.

Tissue	Species	Condition	Article Type	Year	Citation
Sports Injuries	Humans	Several	Review	2022	Herdea et al. [100]
	Humans	Several	Review	2009	Sanchez et al. [101]
MSK	Canine	Several	Review	2021	Sharun et al. [102]
Cartilage	Human	OA	Review	2022	Cash et al. [103]
	Human	Knee OA	Review	2022	Sax et al. [104]
	Human	OA	Review	2022	Trams et al. [105]
	Human	OA	Review	2020	Kydd & Hart [86]
	Human	Defects	Trial	2022	Venosa et al. [106]
Tendons	Human	Epicondylitis	Review	2022	Li et al. [107]
	Human	Tendinopathy	Review	2022	Barman et al. [108]
	Human	Tendinopathy	Review	2022	Cash et al. [103]
	Human	Tendinosis	Trial	2006	Mishra & Pavelko [109]
Ligaments	Porcine	ACL	Trial	2007	Murray et al. [110]
	Human	ACL	Review	2013	Braun et al. [111]
	Human	ACL/MCL	Review	2022	Kunze et al. [97]
IVD	Human	Degeneration	Review	2020	Chang et al. [112]
	Animal	Degeneration	Review	2017	Li et al. [113]
	Human	Degeneration	Review	2017	Basso et al. [114]
	Human	Low Back Pain	Trial	2022	Akeda et al. [115]
Menisci	Human	Sports	Review	2022	Herdea et al. [100]

OA = Osteoarthritis; ACL = Anterior cruciate ligament; MCL = Medial collateral ligament; IVD = intervertebral disc. Citations are representative of the field and many more exist in PubMed for some categories.

**Table 2 biomedicines-10-01570-t002:** Applications of Mesenchymal Stem/Progenitor Cells for Connective Tissue Repair and Regeneration.

Tissue	Species	Condition	Article Type	Year	Citation
Orthopedic Disease	Humans	Several	Review	2022	Malekpour et al. [138]
	Humans	Several	Review	2022	Ren et al. [139]
	Horses	Lameness	Original	2019	Longhini et al. [140]
Cartilage/OA	Human	General	Review	2021	Zha et al. [141]
	Human	General	Review	2021	Vahedi et al. [142]
	Human	Defects	Review	2021	Meng et al. [143]
	Human	Defects	Trial	2018	Shimomura et al. [137]
Tendons	Preclinical	General	Review	2016	Leong & Sun [144]
	General	Injury	Review	2021	Liu et al. [145]
	Human	Tendinopathy	Review	2021	Meeremans et al. [146]
Ligaments	Preclinical	ACL	Review	2015	Jang et al. [147]
	Human	ACL	Review	2015	Jang et al. [147]
Menisci	Preclinical	Injury	Review	2015	Yu et al. [148]
	Human	Injury	Review	2017	Chew et al. [149]
	All	Injury	Review	2021	Rhim et al. [150]
	All	Injury	Review	2022	Zhou et al. [151]
IVD	All	Degenerated	Review	2021	Croft et al. [152]
	All	Degenerated	Review	2022	Liang et al. [153]
	All	Degeneration	Review	2022	DiStefano et al. [154]
Muscle	Rat	Injury	Original	2021	Barbon et al. [155]

OA = Osteoarthritis; ACL = Anterior cruciate ligament; IVD = Intervertebral disc. Citations are representative of the field and many more exist in PubMed.

## Data Availability

N/A No original data was presented in this article.

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
