# Peer review of "Creating an Optimal In Vivo Environment to Enhance Outcomes Using Cell Therapy to Repair/Regenerate Injured Tissues of the Musculoskeletal System"

_biomedicines, 2022, doi:10.3390/biomedicines10071570_

Round 1

Reviewer 1 Report

Summary: The manuscript presents a comprehensive review on outcomes of cell therapy approaches including mesenchymal stem cells and platelet rich plasma from a variety of tissue sources to repair/regenerate injured tissues of the musculoskeletal (MSK) system. Review emphasizes the role of in vivo injury environment in enhancing clinical outcome.  Authors suggest adopt precision medicine approach to optimize tissue repair and regeneration following MSK injuries.

General Comments:  the review presented provides a comprehensive information on how the field of cell/biologic therapies has advanced and what are the unmet challenges as related to MSK System injury repair/regeneration.  Overall, the manuscript is well organized and presented. However, there are no figures/tables for easier visualization of the subject matter and to appreciate the current status of the field.  Continuous text information paragraph after paragraph with no illustrations/figures loses readership attention/enthusiasm and the manuscript in its current format is less attractive.  Please see specific comments for suggestions.

Specific comments:

  1. MSC/PRP therapies as related to MSK repair/regeneration from historical perspective: insert a schematic (line diagram) with timeline to present advancements in cell therapy and cite key references.
  1. Musculoskeletal system tissues: present an illustration of the musculoskeletal system including cartilage, menisci, ligaments, tendons, muscles, bones. Point out areas of injury, repair and regeneration sites with cell therapy.
  1. Insert a table to include period/s of MSC/PRP therapy development (Year/s), type of injury, type of repair, advancements in cell therapy delivery systems, outcomes, references etc.
  2. Include color illustrations/schematics for contents under headlines: 1.2. The Inflammatory response, 5. Enhancing Repair/Regeneration of Injured Connective Tissues of the MSK system, and 6. Use of Single cell preparations of MSC etc., focusing on the “Molecular and Cellular Mechanisms  of Inflammation and Regeneration”.
  3. Suggested to shorten the Title g. Enhancing cell therapy approaches to repair/regenerate injured tissues of the MSK system: Creating an optimal In vivo Environment

Author Response

Please see attached file for our responses to the comments and suggestions

Reviewer 2 Report

This narrative review is based on an intriguing research topic. However, to work on such a wide subject and come out with a concise article, the authors must have a specific topic (or research question), which they clearly do not. The introduction fails to explain why this review was put together, and why the researcher must read this. This lack of focus makes the article extremely hard to follow for the reader.

There is too much information in this review about the injuries on the musculoskeletal tissues, and why they do (or do not) heal, or how they heal. The paragraphs and sections dedicated to these topics are excessively long, which can easily lose the readers' attention

Another very important flaw in the article is with the use of English. There are too many repeating words (thus, therefore, ...). The sentences are very long, and lose their meaning many times. In a narrative review, such lack of clarity can not be allowed.

Author Response

(The authors gave the same response as above.)

Reviewer 3 Report

This review manuscript of David A. Hart and Norimasa Nakamura, titled “Creating an Optimal In Vivo Environment May be Critical for Enhancing Outcomes Using Cell Therapy Approaches to Repair/Regenerate Injured Tissues of the Musculoskeletal System: Controlling inflammation and generating an anabolic environmenty”, covers an upcoming and important field, with fertile ground for development and advancement.

The authors provide an adequate, well written and informative introduction to the article, giving a proper background to the consecutive segments of the manuscript focused literature and existing data.

In a few parts Authors, presented summary of inflammation role in the damage and reconstruction of musculoskeletal system, role of mechanics in connective tissue repair, how to enhance regeneration of injured connective tissue, and finally use of MSC in treatment of tissues. The structure of the paper is written well and present very interesting point of view.

I find the inclusion of the entire musculoskeletal system to be very valuable as opposed to focusing on one narrow topic without considering the impact of the entire environment.

However, I have a few comments:

The title of the manuscript and especially its length is unacceptable! The authors should shorten it at least twice.

This paper also lacks a section/paragraph on unauthorized medical experiments using MSCs, lack of clear legal regulations and medical problems arising from the use of MSCs as a drug without sufficient medical data.

Minor corrections: Spelling errors throughout the manuscript should be revised.

In short, this paper should accepted after minor correction.

Author Response

(The authors gave the same response as above.)

Reviewer 4 Report

The authors presented an interesting review on the possible effect of the inflammatory microenvironment on the regenerative outcomes of mesenchymal stem cell-based therapies in case of musculoskeletal (MSK) tissue injuries, suggesting that controlling inflammation and enhancing the anabolic environment could improve the recovery process.

The review discusses a current and fascinating topic which was clearly and concisely presented. The text is well organized and quite well written, but some minor check of English grammar needs to be performed.

Some major concerns to be addressed are following listed:

1) A general introduction should be added before directly getting to the heart of the topic, also describing the principal aim of the review article and the specific issues that it is going to discuss.

2) A paragraph describing the methods/criteria applied for literature search is missing. It would be preferable to add a methodology section.

3) In Paragraph 5, when presenting platelet-rich hemocomponents as an approach to enhance regenerative processes of the injured MSK tissues, recent findings are worth mentioning about the possibility to concentrate circulating multipotent stem cells within these products, and consequent implications for regenerative therapies (i.e., Caloprisco et al. Transfus Apher Sci. 2010;42:117-24; Di Liddo et al. J Cell Mol Med. 2018;22:1840-1854; Barbon et al. J Tissue Eng Regen Med. 2018;12:1891-1906).

4) Related to the previous point, circulating multipotent cells isolated from peripheral blood (see for example: Lin et al. J Orthop Translat. 2019;19:18-28; Barbon et al. Tissue Eng Regen Med. 2021;18:411-427; Di Liddo et al. Int J Nanomedicine. 2016;11:5041-5055) could represent a valid therapeutic source to control inflammation in an inflammatory microenvironment, since they can be easily administrated to the patient as an autologous transplant which can locally exert immunomodulatory activity (i.e., Longhini et al. PLoS One. 2019;14:e0212642) within the MSK damage site. These considerations could be added, for example, in Paragraph 6.

5) The review should contain some figures/schemes/tables, which should be referred to within the body of the manuscript.

Author Response

Reviewer #4: (Major Revision)
Point #1: A general introduction should be added before directly getting to the heart of the topic,
also describing the principal aim of the review article and the specific issues that it is going to
discuss.

A new paragraph has been added to the beginning of the review to indicate the issues and
suggestions raised by the Reviewer. We hope this paragraph addresses the concerns raised.

Point #2: A paragraph describing the methods /criteria applied for literature search is missing. It
would be preferable to add a methodology section.

We thank the Reviewer for this suggestion. We have added a sentence to that initial paragraph
mentioned in the response to Point #1 that indicates that the PubMed database was searched for
relevant articles that addressed some of the issues discussed. In some respects, this review is not
a complete assessment of the whole field, but one that is addressing some of the aspects of the
field that are likely contributing to a lack of success in moving the use of cell therapies to
enhance repair of injured MSK tissues. The topic area is quite large, and as indicated by
Reviewer #3, we tried to be both comprehensive and focused. Therefore, we did not feel that a
whole paragraph was needed to address the methodology used for the review. Furthermore, the
authors have >15 years of experience in this area and therefore have built on that experience to
address gaps in the field of cell therapies to repair MSK tissues.

Point #3: In Paragraph 5, when presenting platelet-rich hemocomponents as an approach to
enhance regenerative processes of the injured MSK tissues, recent findings are worth mentioning

about the possibility to concentrate circulating multipotent stem cells within these products, and
consequent implications for regenerative therapies.

Thank you for this suggestion. We have downloaded and read the three references you
mentioned. Certainly, it has been known that some MSC-like cells do circulate in the blood,
and that the numbers decline with aging. However, the numbers are quite low and often require
considerable expansion in vitro to obtain sufficient numbers to use in studies. The three reports
indicated demonstrate that such cells are present, but they were not applied to repairing MSK
injuries. Thus, while the reports are interesting, they likely do not fall within the topic area at the
present time but could in the future. Therefore, we respectfully declined to include a discussion
regarding these studies in the revised manuscript.

Point #4: Related to the previous point, circulating multipotent cells isolated from peripheral
blood (see for example Lin et al; Barbon et al; Di Litto et al) could represent a valid therapeutic
source to control inflammation in an inflammatory microenvironment, since they can easily be
administered to the patient as an autologous transplant which can locally exert
immunomodulatory activity (i.e. Longhini et al) within the MSK damage site. These
considerations could be added, for example, in Paragraph 6.

Thank you for these suggestions. We have downloaded the indicated papers and read them over
carefully. The Lin paper described a new protocol for such cells isolated from peripheral blood,
and the Di Liddo paper described studies looking at nanopatterned acellular value conduits and
peripheral blood multipotent cells. While both of these are interesting reports, from our
assessment, they do not align well with the focus of the present review and therefore, we have
respectfully declined to include them in the revised manuscript.

In contrast, the Barbon et al (2021) and Longhini et al (2019) reports used circulating stem cells
to assess their function in an in vivo rat model of muscle injury and in lame horses, respectively.
Therefore, these two studies/reports did align with the focus of this review and were
incorporated into new Table 2.

Point #5: The review should contain some figures/schemes/tables, which should be referred to
within the body of the manuscript.

Thank you for these suggestions. They were also made by other reviewers and therefore, we have
now included Figure 1 and Tables 1 & 2 into the revised manuscript.

We thank the Reviewer’s for their time and suggestions and look forward to receiving notice
regarding the suitability of our review for publication in Biomedicines.

With Best Regards,

Round 2

Reviewer 1 Report

Authors have addressed review concerns in part.  No further questions.

Reviewer 4 Report

The authors responded adequately to the criticisms raised. The work can be published in the present form.